# Qualitative and Quantitative Analysis of *C*-glycosyl-flavones of *Iris lactea* Leaves by Liquid Chromatography/Tandem Mass Spectrometry

**DOI:** 10.3390/molecules23123359

**Published:** 2018-12-18

**Authors:** Dan Chen, Yu Meng, Yan Zhu, Gang Wu, Jun Yuan, Minjian Qin, Guoyong Xie

**Affiliations:** 1Department of Resources Science of Traditional Chinese Medicines, School of Traditional Chinese Pharmacy, China Pharmaceutical University, #24 Tongjiaxiang, Gulou District, Nanjing 210009, China; 1721020319@stu.cpu.edu.cn (D.C.); mengyu19900616@163.com (Y.M.); cpuzy@126.com (Y.Z.); woosmail@163.com (G.W.); 2Jiangsu Key Laboratory of Regional Resource Exploitation and Medicinal Research, Huaiyin Institute of Technology, Huai’an 223003, Jiangsu, China; yuanjun1109@126.com

**Keywords:** *Iris lactea* Pall. var. *chinensis* (Fisch.) Koidz., HPLC-Q-TOF-MS/MS, qualitative analysis, quantitative analysis, *C*-glycosylflavone

## Abstract

*Iris lactea* Pall. var. *chinensis* (Fisch.) Koidz. is a traditional medicinal plant resource. To make full use of the *I. lactea* plant resources, constituents of *I. lactea* leaves were determined by high performance liquid chromatography (HPLC)-quadrupole time-of-flight tandem mass spectrometry and 22 *C*-glycosylflavones were identified or tentatively identified. Optimal extraction of *I. lactea* leaves was established via single factor investigations combined with response surface methodology. Then, HPLC coupled with a diode array detector was used to quantitatively analyze the six main components of 14 batches of *I. lactea* leaves grown in different areas. The results showed the *C*-glycosylflavones were the main components of *I. lactea* leaves, and the total contents of detected components were relatively stable for the majority of samples. These results provide a foundation for the development and utilization of *I. lactea* leaves.

## 1. Introduction

*Iris lactea* Pall. var. *chinensis* (Fisch.) Koidz. is a perennial herb of the Iridaceae family. This plant is widely distributed in China and was first recorded in *Shen Nong’s Herbal Classic*. The seeds, flowers and roots are used as a folk medicine for the treatment of jaundice, pharyngitis, hemorrhoids, ulcer, vomiting blood and stranguria with turbid discharge, and the leaves are used to treat pharyngitis and joint pain of the lower back and legs [1]. Modern research has shown that *I. lactea* contains flavonoids, benzoquinones, stilbenes and volatiles, and possesses various bioactivities, including anti-inflammatory, antioxidant, anti-tumor, and anti-radiation effects [2,3,4,5,6,7,8,9]. In particular, irisquinone which is isolated from *I. lactea* seeds, has been successfully used for lung cancer, esophageal cancer, head and neck cancer as an antineoplastic agent and radiosensitizer [10]. In recent years, research on the composition and bioactivity of *I. lactea* has concentrated on the seeds and rhizomes, but seldom on its leaves.

Leaves are the main part of *I. lactea*, representing abundant biomass, and aside from their medicinal value, they are also a type of pasture in the absence of winter forage [11,12]. In our previous studies, a series of *C*-glycosylflavones which possessed anti-inflammatory and cytotoxicity activities were isolated from *I. lactea* leaves [11,13]. The activities are beneficial for people and animals, and meet the requirements of the development and utilization of these medicinal plant resources.

High performance liquid chromatography (HPLC) equipped with quadrupole time-of-flight tandem mass spectrometry (Q-TOF-MS/MS) has become an essential analytical tool in the modernization of Traditional Chinese Medicine. The method is efficient and rapid at determining the molecular weight and characteristic fragment ions, by which the structure of compounds can be identified quickly [14,15,16,17]. In the study, we used HPLC-Q-TOF-MS/MS to systematically separate and identify the compounds in *I. lactea* leaf extracts. Subsequently, HPLC coupled with a diode array detector (HPLC-DAD) was used for quantitative analysis of six main components of *I. lactea* leaves from different growing areas. This study provides a valid approach to the comprehensive quality-evaluation and better utilization of *I. lactea* leaves.

## 2. Results and Discussion

### 2.1. Compound Identifications

The chromatograms and total ion chromatograms of standards and samples of *I. lactea* are displayed in Figure 1 and each peak in chromatograms is numbered with a number corresponding to the compound information listed in Table 1. Twenty-two chemical constituents were identified or tentatively identified from *I. lactea* leaves based on their retention time, maximum UV absorption, mass spectum and relevant literature [11,18,19,20,21,22,23,24,25]. The chemical structures of the compounds are shown in Appendix A.

Compounds **A1**, **A3** and **A4** possessed similar maximum absorptions of about 239 (shoulder peak), 260 (or 257), 320 and 360 nm, which are characteristic UV features of xanthones. The fragment ions of compound **A3** showed 331 [M − H − 90]^−^, 301 [M − H − 120]^−^ and 271 [M − H − 150]^−^, which are typical of *C*-glucosides [18]. By comparing with mass spectra of a reference standard and previously reported data [18], **A3** was identified as mangiferin. By a similar method, compounds **A1** and **A4** were tentatively identified as neomangiferin and isomangiferin, respectively [18].

Compounds **A2** and **A5**–**A22** showed similar UV spectra with absorption maxima at 240–280 and 320–360 nm, and a similar fragmentation pattern, which showed successive losses of 60, 90 and 120 Da, which is typical of flavone *C*-glucosides [18]. Compound **A2** (*m/z* 593.1514 [M − H]^−^) exhibited UV absorption peaks at 268 and 320 nm and the molecular formula C_27_H_30_O_15_, which indicated that it was a flavone; fragment ions at *m/z* 503 [M − H − 90]^−^ and 473 [M − H − 120]^−^ indicated that **A2** was a flavone *C*-glucoside, and the fragment ions at 341 [M − H − 90 − 162]^−^ and 311 [M − H − 120 − 162]^−^ showed that it was also a *O*-glucoside, as did the fragment ion at *m/z* 119 and related references [19,20]. Compound **A2** was thus tentatively identified as apigenin-7-*O*-glucoside-6-*C*-glucoside. By a similar method, compounds **A5**–**A9** were tentatively identified as luteolin-6-*C*-*β*-d-glucoside, swertiajaponin, saponaretin, scoparin and swertisin-2″-*O*-rhamnoside-4′-*O*-glucoside, respectively [19,21,22]. Compound **A10** exhibited the same fragmentation pathway as **A9**, but had a higher molecular weight (42 Da); using information from the literature [22], **A10** was tentatively identified as swertisin-2′′-*O*-(4′′′-acetylrhamnoside)-4′-*O*-glucoside.

Compound **A11** showed a molecular ion at *m/z* 605.1905 [M − H]^−^, and fragment ions at *m/z* 459 [M − H − 146]^−^ and 339 [M − H − 120 − 146]^−^, which indicated that it was a *O*-rhamnoside. The fragment ions at *m/z* 485 [M − H − 120]^−^, 441 [M − H − 146 − 18]^−^, 381 [M − H − 146 − 18 − 60]^−^, 351 [M − H − 146 − 18 − 90]^−^ and 321 [M − H − 146 − 18 − 120]^−^ showed that **A11** was a *C*-glucoside; in addition, it showed other fragment ions at *m/z* 307, 163 and 103. By comparing an authentic standard and the corresponding UV and MS data with literature values [23], **A11** was unambiguously identified as embinin. Compounds **A18** and **A22** showed a similar fragmentation pathway to, but possessed one or two more acetyl groups than compound **A11**. By comparison with authentic standards and literature data [11,24], compounds **A18** and **A22** were identified as 4′′′-acetyl-embinin and embinin C, respectively. Compounds **A14**, **A16** and **A21** were isomers of irislactin C, which showed a similar fragmentation pathway to **A18**. The main differences in these compounds were the different substitutions of the acetyl groups. Combined with the molecular weight, retention time and literature [10,24,25], compounds **A14, A16** and **A21** were tentatively identified as 2′′′-acetyl-embinin, 3′′′-acetyl-embinin and irislactin B, respectively.

Compound **A15** had a molecular ion at *m/z* 883.2489 [M + HCOO]^−^, a similar fragmentation pattern to compound **A11** and a base peak at *m/z* 633 [M − H − 162 − 42]^−^; it also had a fragment ion at *m/z* 675 [M − H − 162]^−^ which was not tested, so we speculated that **A15** possessed a glucoside residue connected with an acetyl group. Fragment ions at *m/z* 717 [M − H − 120]^−^, 513 [M − H − 162 − 42 − 120]^−^, 427 [M − H − 162 − 42 − 42 − 146 − 18]^−^ and 307 [M − H − 162 − 42 − 42 − 146 − 18 − 120]^−^ were found in **A15**. By comparing the authentic standards and their corresponding UV and MS data with literature values [11], **A15** was unambiguously identified as irislactin C. Compounds **A12**, **A13** and **A17** possess the same molecular formula as compound **A15**, and showed a similar fragmentation pathway to **A15**, therefore Compounds **A12**, **A13** and **A17** were tentatively identified as the isomers of irislactin C. Compound **A19** showed a molecular ion at *m/z* 925.2628 [M + HCOO]^−^, and possessed the same pathway as **A15**. By comparing the molecular weight, authentic standards and their corresponding UV and MS data with literature values [25], **A19** was unambiguously identified as irislactin A. Compound **A20** showed the same molecular formula and a similar fragmentation pathway to **A19**, Thus compound **A20** was tentatively identified as an isomer of irislactin A.

Twenty-two compounds including three xanthones and nineteen flavones were thus identified or tentatively identified from *I. lactea* leaves. All constituents identified were *C*-glycosylflavones, including twelve acetylated *C*-glycosylflavones. The literature reports indicate that *C*-glycosylflavones are widely distributed in plant kingdom, and found in algae, bryophytes, ferns, gymnosperms and angiosperms, involving hundreds of species of plants from different families and genera, such as Characeae, Conocephalaceae, Psilotaceae, Cycadaceae and Compositae, etc [26]. These kinds of ingredients show various pharmacological activities, including anti-oxidant [27], anti-inflammatory [28], anti-diabetes [29], anti-tumor [30], anti-virus [31], cardiovascular protection [32], liver-protection [33] and memory amelioration [34]. Among the compounds identified from *I. lactea* leaves, mangiferin showed good anti-inflammatory, anti-diabetes, and anti-tumor pharmacological activity, and is one of the hotspots in current studies [35,36], while acetylated *C*-glycosylflavones showed poor activity in the literature [11,25]. On the whole, the *C*-glycosylflavones are worthy of further study.

### 2.2. Optimization of the Extraction Process

When the degree of comminution reached 80-mesh, the extraction ratio increased slowly (Appendix A). Thus, 80-mesh was chosen as one of the optimal extraction parameters after considering the centrifugation, filtration and other experimental factors. The total peak area of target components for three different extraction methods showed no significant difference (Appendix A). However, ultrasound extraction was finally chosen for optimization because the methods of soaking and hot reflux were more operation-complex and time-consuming. The extraction efficiency of methanol was higher than that of ethanol at the same concentration (Appendix A). Moreover, with increasing solvent concentration, the extraction efficiency initially increased and then decreased. Therefore, 40–80% methanol solution was selected as solvent range for response surface design [37,38]. In the investigation of liquid-solid ratio, extraction efficiency improved with the increase of liquid volume but with no obvious difference between 20 and 25 mL of methanol (Appendix A). For reasons of experimental cost, the liquid-solid ratio of 1:15–1:25 was chosen for response surface optimization. In addition, with the increase of extraction time, the total peak area of target components rose progressively more slowly (Appendix A). Consequently, extraction time of 15–45 min was selected as the level of response surface design. In assessment of extraction frequency, the total peak area presented an increasing trend, but the efficiency of three extractions was almost the same as that for two (Appendix A). Hence, the frequency of two extractions was chosen for optimization [39].

Subsequently, the extraction parameters were further optimized by Box-Behnken design experiment. The data displayed in Appendix A were fitted to a quadratic polynomial model using response surface methodology.

The obtained encoding equation was as follows:Y = 3.75 + 0.24 A + 0.066 B + 0.060 C − 0.010 AB − 0.018 AC + 0.17 BC − 0.40 A^2^ − 0.11 B^2^ − 0.11 C^2^(1)
and the true-value equation was as follows:Y = −2.00075 + 1.34 A + 0.13 B − 8.23 × 10^−3^ C − 1.0 × 10^−4^ AB − 5.83 × 10^−5^ AC + 2.23× 10^−3^ BC − 9.9 × 10^−4^ A^2^ − 4.44 × 10^−3^ B^2^ − 4.82 × 10^−4^ C^2^(2)
where Y is the extraction efficiency of the main active components in *I. lactea* leaves (shown by the total peak area of six main components), and variables A, B and C represent the methanol concentration (%), liquid-solid ratio (mL·(0.5 g)^−^^1^)and extraction time, respectively.

To verify the feasibility of the regression equation, significance (α = 0.05) of the model and coefficient was tested (Appendix A). The *p* value (<0.0001) and correlation coefficient (R^2^ = 0.9977) of the model demonstrated the extreme significance of the regression model and linear relationship between Y and the dependent variable. Additionally, the lack of fit (*p* = 0.0759 > 0.05) also suggested that this equation had a good fit and little deviation for corresponding true values. Thus, this model could be used to adequately evaluate the experimental results. Because the *p* values of the regression coefficients of variables (A, B and C), as well as their interaction (BC) and quadratic effects (A^2^, B^2^ and C^2^) were less than 0.0001, this implied that they significantly affected the *Y* value, but their interactions (AB and AC) did not (*p* values of 0.3845 and 0.1486, respectively, i.e., >0.05).

Subsequently, the 3D response surface and the corresponding 2D contour map (Appendix A) were used to further analyze the factor interactions, where the steeper the curve is, the greater effect the factor has on the response value. When methanol concentration was constant, the liquid-solid ratio had no obvious influence on extraction efficiency; when the liquid-solid ratio was constant, the methanol concentration initially increased and then decreased (Appendix A). The extraction efficiency changed gently with time, increasing with rising methanol concentration up to a certain value and subsequently decreasing (Appendix A). The combined influence of liquid-solid ratio and extraction time had a slight impact on extraction efficiency (Appendix A). In summary, the optimal conditions for maximum response values, calculated via Design-Expert, were methanol concentration of 65.16%, liquid-solid ratio of 25.73:1 and extraction time of 47.07 min. For convenience and less cost, the corresponding optimum values were 65%, 25:1 and 47 min, for which the true value was only 2% lower than the predicted value.

### 2.3. Optimization of Chromatographic Conditions

To optimize the chromatographic separation efficiency, several influence factors of detection wavelengths (254 nm and 270 nm), mobile phase (methanol/(acid) water and acetonitrile/(acid) water), column temperature (25 °C, 30 °C and 35 °C), flow rate (0.8 mL·min^−1^ and 1 mL·min^−1^) and injection volume (10 μL, 15 μL and 20 μL) were tested. The optimized parameters were selected as mobile phases of 0.1% formic acid-water (A) and acetonitrile (B), flow rate of 0.8 mL·min^−1^, column temperature of 30 °C, injection volume of 15 μL, detection wavelength of 270 nm and program run time of 45 min after comparing the peak shape and analysis time.

### 2.4. Method Validation

#### 2.4.1. Linearity and Limits of Detection (LOD) and Quantitation (LOQ)

The calibration curves of the six reference compounds of mangiferin, embinin, irislactin C, irislactin A, embinin A and embinin C were drawn using the results of determination (Appendix A). The calibration curve, correlation coefficient, linear range, LOD and LOQ of each reference compound were obtained (Table 2), and the reference compounds both showed a good linear relationship (R^2^ ≥ 0.9998) within the test ranges.

#### 2.4.2. Precision, Repeatability, Stability and Recovery

The relative standard deviations (RSDs) of intra- and inter-day precision, repeatability and stability investigation of mangiferin, embinin, irislactin C, irislactin A, embinin A and embinin C were all <2%, indicating that our method had good precision, repeatability and stability (Appendix A). Additionally, the recovery range of 97–101% (RSD < 3%) indicated high recovery and reliability (Appendix A).

### 2.5. Quantitative Analysis of HPLC-DAD for Flavonoids of I. lactea Leaves

Using the chromatograms, the six main components of *I. lactea* leaves from different regions were quantitatively analyzed, where the variation ranges of mangiferin, embinin, irislactin C, irislactin A, embinin A and embinin C were 0.48–2.16, 0.88–11.78, 0.75–5.56, 0.77–3.11, 0.92–6.67 and 0.49–12.38 mg·g^−1^, respectively (Table 3). The contents of mangiferin and irislactin A varied narrowly, but those of embinin, embinin A, irislactin C and embinin C varied widely; the results indicated that the contents of tested compounds of different samples showed certain differences. The total contents of six main components in the samples from Nanjing (S3), Tianjin (S6) and Haidian, Beijing (S10) had the higher content (>20 mg·g^−1^), samples from Liaoning (S14) had the lowest content (<15 mg·g^−1^) and the majority of samples were a relatively stable. (15–20 mg·g^−1^) (Figure 2).

In addition, even in the same area, the total contents of six main components in leaves of different batches were statistically different (Table 3 and Table 4). For instance, in *I. lactea* leaves collected from Nanjing, Jiangsu, the total content of six main components in smple S3 was higher than that of samples S1 and S2. Similarly, the total content of sample S10, gathered from Haidian, Beijing, was higher than that of sample S9 from Dongcheng District, Beijing. The total content of sample S6 (Tianjin) was higher than that of sample S5 (Jixian County, Tianjin). The reason for the differences in total contents is likely such factors as geographical location of sampling such as sample S3 and S13, sample S10 and S11, the phenological influence such as sample S1–S3, samples S13 and S14, and chemical transformations among compounds. In previous study, we found the phenomenon that some compounds had mutual transitions, such as irislactin A and embinin C [11]. In addition to the above factors, there may be other factors affecting the change of chemical composition content, which needs to be further studied and analyzed.

## 3. Materials and Methods

### 3.1. Chemicals and Plant Material

The standards of mangiferin, embinin, irislactin C, irislactin A, embinin A and embinin C were made in our laboratory. The purity of each compound was determined to be higher than 96% by NMR, MS and area normalization method. Chromatographic grade methanol and formic acid were purchased from Nanjing Chemical Reagents Co. Ltd. (Nanjing, China). Acetonitrile was purchased from Merck (Darmstadt, Germany). Wahaha pure water was obtained from Hangzhou Wahaha Group Co. Ltd. (Hangzhou, China). Sample information for *I. lactea* leaves is shown in Table 4.

### 3.2. Preparation of Samples and Standard Solutions

The sample solution was prepared by extracting the powder of *I. lactea* leaves (accurately weighed 0.50 g) in 20 mL of 70% methanol. Then, supernatant volume was amalgamated and shaken in a 50-mL volumetric flask after two ultrasonic extractions at 25 °C, 100 W for 45 min (Kunshan Wo Chuang Ultrasonic Instrument Co. Ltd., Kunshan, China) and centrifuged for 20 min at 12000 rpm. Subsequently, the solution was stored in a refrigerator at 4 °C and filtered through a 0.22 μm membrane (Tianjin Xinxian Technology Co. Ltd., Tianjin, China) filter until analysis.

The standards of mangiferin (0.22 mg), embinin (1.27 mg), irislactin C (2.10 mg), irislactin A (2.10 mg), embinin A (1.12 mg) and embinin C (2.50 mg) were accurately weighed. Next, they were individually dissolved in a 2-mL volumetric flask in methanol. Each standard solution was obtained after adjusting to a constant volume. The mixed standard solution was obtained by appropriately mixing each standard solution in a 2-mL volumetric flask for mangiferin (0.02244 mg·mL^−1^), embinin (0.198 mg·mL^−1^), irislactin C (0.100 mg·mL^−1^), embinin A (0.1134 mg·mL^−1^), irislactin A (0.0371 mg·mL^−1^), embinin C (0.180 mg·mL^−1^). These were stored in a refrigerator at 4 °C and filtered through a 0.22 μm membrane filter until analysis.

### 3.3. Qualitative Analysis of HPLC-Q-TOF-MS/MS for Chemical Constituents of I. lactea Leaves

Chromatographic analyses were performed using a high performance liquid chromatograph (Agilent Technologies Inc., Santa Clara, CA, USA) coupled to an electrospray ionization (ESI) mass spectrometer (Agilent Technologies Inc., Santa Clara, CA, USA). Chromatographic separation was conducted on an Agilent Zorbax SB-C18 column (3.0 mm × 150 mm, 3.5 μm). The mobile phases consisted of 0.1% formic acid–water (A) and acetonitrile (B), and the gradient elution program was set as follows: 0 min, 5% B; 5–10 min, 11% B; 15 min, 19% B; 20–24 min, 24% B; 25 min, 27% B; 28 min, 35% B; 30 min, 38% B and 35 min, 70% B. The flow rate was 0.8 mL·min^−1^, injection volume was 15 μL, column temperature was 30 °C and detection wavelength was 270 nm. The ESI was applied in negative ion modes for mass analysis and detection. The optimized parameters were as follows: capillary voltage, 3000 V; conical-hole voltage, 60 V; nebulizing-gas pressure, 35 psi; drying-gas flow rate, 10 L·min^−1^; drying-gas temperature, 320 °C; and mass spectral range, *m/z* 100–2000.

### 3.4. Optimization of the Extraction Process

#### 3.4.1. Single Factor Experiments

Single factor tests were carried out to optimize the flavonoid extraction. The extraction conditions showed as follows. Powder of *I. lactea* leaves (0.5 g, Sample S6) was used. Comminution degree (20, 40, 60, 80 and 100 mesh), extraction method (soak for 12 h, ultrasonication for 30 min at room temperature and reflux 1 h at 80 °C), methanol/ethanol concentration (40, 60, 80 and 100%), liquid–solid ratio (10, 15, 20 and 25 mL of methanol), extraction time (15, 30, 45 and 60 min) and frequency (1, 2 and 3) were investigated, respectively. Each level was run in triplicate. When one of the factors was experimented, conditions of other factors were the same as “3.2 Preparation of Samples”. The optimal extraction conditions were preliminarily chosen according to total contents of mangiferin, embinin, irislactin C, irislactin A, embinin A and embinin C determined by HPLC.

#### 3.4.2. Box-Behnken Response-Surface Design Experiment

Box-Behnken design conducted using Design-Expert software (version 8.0.6, Stat-Ease Inc., Minneapolis, MN, USA) (Appendix A) was chosen for optimized extraction of flavonoids in *I. lactea* leaves based on results of single factor experiments. Since it is much more efficient than the three-level full factorial designs [40]. Each factor was set as the following levels: methanol concentration (40, 60 and 80%) (A), liquid-solid ratio (15:1, 20:1 and 25:1) (B) and extraction time (15, 30 and 45 min) (C).

### 3.5. Method Validation

#### 3.5.1. Preparation of Sample Solution

The sample solution was prepared based on the result of the Box-Behnken response-surface design experiment. Powder of *I. lactea* leaves (0.5 g, sample S6) was accurately weighed and extracted in 25 mL of 65% methanol. Subsequently, supernatant volume was amalgamated and shaken in a 50 mL volumetric flask after twice ultrasonic extractions and centrifuged for 20 min at 12000 rpm. The extract was stored in a refrigerator at 4 °C and filtered through a 0.22 μm membrane filter until analysis.

#### 3.5.2. Linearity, LOD and LOQ

The mixed standard solutions in nine different concentrations were prepared by gradient dilution with methanol prior to analysis using HPLC. The least squares method was used for regression analysis, with injection concentration (mg·mL^−1^) as the abscissa and peak area of the index components as the ordinate. The mixed standard solution was diluted by methanol to determine the LOD and LOQ. The concentrations when the ratios of sign-to-noise were 3:1 and 10:1 were selected as the LOD and LOQ, respectively.

#### 3.5.3. Precision, Repeatability, Stability and Recovery

Five repeated injections of the mixed standard solution in the same day and three repeated injections per day for three consecutive days were used to evaluate of intra- and inter-day precision, respectively. Six sample solutions were prepared independently to check repeatability. The sample solution was injected at 0, 4, 8, 12, 24, 48 and 72 h separately for analysis of stability. To investigate recovery, six sample solutions prepared by adding mixed standard solution to 0.25 g of *I. lactea* leaves (S6) were analyzed.

### 3.6. Quantitative Analysis of HPLC-DAD for Flavonoids of I. lactea Leaves

Quantitative analysis of six main components of *I. lactea* leaves from different producing areas was performed individually by HPLC-DAD based on the optimum extract parameters. Contents of six components in different samples were calculated via linear regression equation.

### 3.7. Data Analysis

All data were collected and analyzed using Masshunter Qualitative Analysis Software B 03.00 ChemStation software (Agilent Technologies Inc., Santa Clara, CA, USA). Data treatment was carried out using Microsoft Excel software (Microsoft Corp., Redmond, WA, USA) and IBM SPSS software 22.0 (IBM Corp., Armonk, NY, USA).

## 4. Conclusions

HPLC-Q-TOF-MS/MS was used to qualitatively analyze the constituents of *I. lactea* leaves, and 22 *C*-glycosylflavones were identified or tentatively identified. If a more detailed classification is desired, compounds **A1**, **A4** and **A5** belong to the xanthone *C*-glycosides, and the other compounds are flavone *C*-glycosides, especially, compounds **A10** and **A12**–**A22** which belong to the flavone *C*-glycosides with acetyl groups. According to the literatures and our studies [11,25], we found the flavone *C*-glycosides with acetyl groups are the characteristic ingredients of *I. lactea* leaves, and these types of compounds may possess chemotaxonomic significance to distinguish *I. lactea* from the other genera.

After optimizing the extraction method, 14 batches of *I. lactea* leaves gathered from 10 different growing districts in eight Chinese provinces were quantitatively analyzed. The results showed the *C*-glycosylflavones were the main components of *I. lactea* leaves, and the total contents of detected components were relatively stable for the majority of samples. Among them, the samples from Nanjing (sample S3), Tianjin (sample S6) and Haidian, Beijing (sample S10) had the higher content (>20 mg·g^−1^), samples from Liaoning (sample S14) had the lowest content (<15 mg·g^−1^) (Figure 2). This might be caused by geographical location of sampling, phenological information and chemical transformations between compounds. These relevant factors will need to be investigated, analyzed and optimized to improve quality of *I. lactea*.

## Figures and Tables

**Figure 1 molecules-23-03359-f001:**
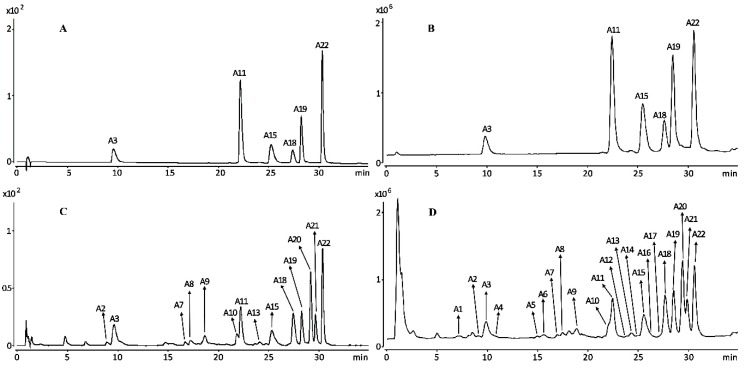
Chromatograms (**A**,**C**) and total ion chromatograms (**B**,**D**) of standards and samples of *Iris lactea* ((**A**,**B**): Standard; (**C**,**D**): Sample). **A3**: mangiferin, **A11**: embinin**, A15**: irislactin C, **A18**: embinin A, **A19**: irislactin A and **A22**: embinin C.

**Figure 2 molecules-23-03359-f002:**
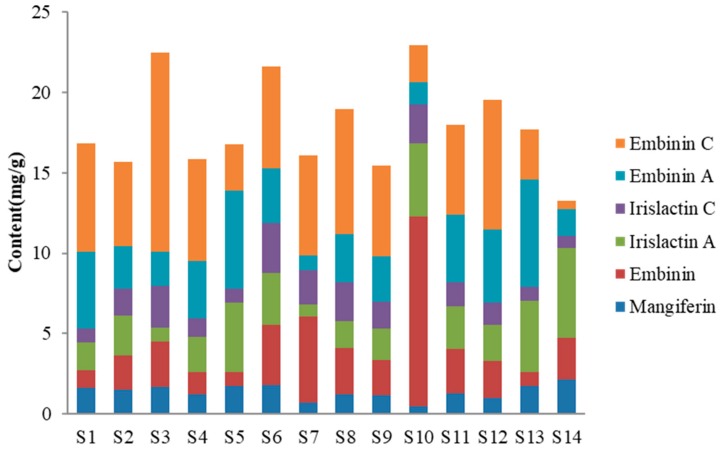
Total contents of main components in *Iris lactea* leaves from different areas.

**Table 1 molecules-23-03359-t001:** Characterization of chemical constituents of *Iris lactea* by HPLC-DAD-Q-TOF-MS/MS.

Compound	Tr (time)	UV (nm)	Quasi-Molecular (Error, ppm)	Molecular Formula	*m*/*z* Calculated	MS/MS Fragments	Proposed Compound	References
**A1**	7.089	239, 257, 320, 360	583.1261 (−2.55) [M − H]^−^	C_25_H_28_O_16_	583.1246	565, 493, 463, 331, 301, 259	neomangiferin	[18]
**A2**	9.188	268, 320	593.1514(−0.3) [M − H]^−^	C_27_H_30_O_15_	593.1512	575, 503, 473, 341, 311, 282, 119	apigenin 7-*O*-glucoside-6*C*-glucoside	[19,20]
**A3**	9.863	239, 260, 320, 360	421.0785 (−2.04) [M − H]^−^	C_19_H_18_O_11_	421.0762	403, 301, 331, 285, 271, 259, 243, 215	mangiferin	[18]
**A4**	10.586	239, 260, 320, 360	421.0779 (−0.53) [M − H]^−^	C_19_H_18_O_11_	421.0776	331, 301, 285, 271, 258, 243, 215	isomangiferin	[18]
**A5**	15.002	268, 352	447.0934 (−2.49) [M − H]^−^	C_21_H_20_O_11_	447.0929	429, 357, 327, 331, 299, 133	luteolin 6-*C-**β*-d-glucoside	[21]
**A6**	15.561	252 (sh*), 272, 318	461.1073 (3.49) [M − H]^−^	C_22_H_22_O_11_	461.1091	446, 313, 298, 285, 133	swertiajaponin	[19]
**A7**	16.899	267, 336	431.0988 (−0.05) [M − H]^−^	C_21_H_20_O_10_	431.0986	341, 323, 311, 283, 117	Saponaretin	[21]
**A8**	17.613	256, 332	461.1079 (2.33) [M − H]^−^	C_22_H_22_O_11_	461.1089	371, 341, 298	scoparin	[19]
**A9**	18.899	270, 324	799.2299(0.49) [M + HCOO]^−^	C_34_H_42_O_19_	754.2320	753, 659, 633, 591, 427, 307	Swertisin 2″-*O*-rhamnoside-4′-*O*-glucoside	[22]
**A10**	22.065	270, 327	841.2448(−4.79) [M + HCOO]^−^	C_36_H_44_O_20_	796.2426	795, 659, 633, 591, 427, 307	Swertisin 2″-*O*-(4′′′-acetylrhamnoside)-4′-*O*-glucoside	[22]
**A11**	22.581	270, 338	605.1905 (−4.81) [M − H]^−^	C_29_H_34_O_14_	605.1876	485, 459, 441, 423, 381, 363, 351, 339, 321, 307, 163, 103	embinin	[23]
**A12**	23.888	270, 332	883.2489(−2.03) [M + HCOO]^−^	C_38_H_46_O_21_	838.2532	837, 675, 633, 555, 513, 427, 307	The isomer of irislactin C	[11]
**A13**	24.163	270, 332	883.2529(1.08) [M + HCOO]^−^	C_38_H_46_O_21_	838.2532	837, 675, 633, 555, 513, 427, 307	The isomer of irislactin C	[11]
**A14**	24.920	268, 332	647.1994(−1.94) [M − H]^−^	C_31_H_36_O_15_	647.1979	605, 587, 459, 441, 381, 339, 145, 101	2′′′-acetyl-embinin	[11,24]
**A15**	25.505	268, 330	883.2489(2.11) [M + HCOO]^−^	C_38_H_46_O_21_	838.2532	837, 633, 513, 427, 307	irislactin C	[11]
**A16**	26.331	270, 330	647.1979(−0.42) [M − H]^−^	C_31_H_36_O_15_	647.1978	605, 527, 459, 381, 351,339 127, 101	3′′′-acetyl-embinin	[11,24]
**A17**	27.260	268, 328	883.2526(−2.51) [M + HCOO]^−^	C_38_H_46_O_21_	838.2532	837, 675, 633, 555, 513, 427, 307	The isomer of irislactin C	[11]
**A18**	27.776	268, 330	647.1960(3.34) [M − H]^−^	C_31_H_36_O_15_	647.1977	605, 587, 459, 441, 381, 339, 145, 101	embinin A	[11,24]
**A19**	28.636	270, 328	925.2577(3.87) [M + HCOO]^−^	C_40_H_48_O_22_	880.2637	879, 675, 633, 555, 427, 307	irislactin A	[25]
**A20**	29.290	268, 330	925.2628(−2.77) [M + HCOO]^−^	C_40_H_48_O_22_	880.2637	879, 675, 633, 427, 307	The isomer of irislactin A	[25]
**A21**	29.857	268, 328	689.2145 (0.17) [M − H]^−^	C_33_H_38_O_16_	689.2146	647, 605, 587, 527, 459, 441, 351, 127, 113	irislactin B	[25]
**A22**	30.700	246, 326	689.2079(−1.02) [M − H]^−^	C_33_H_38_O_16_	689.2074	647, 605, 587, 527, 459, 441, 351, 145, 109	embinin C	[11]

sh*: shoulder peak.

**Table 2 molecules-23-03359-t002:** Calibration curves, liner range, LOD and LOQ of six reference compounds.

Analyte	Calibration Curves	R^2^	Liner Range (µg·mL^−1^)	LOD (ng·mL^−1^)	LOQ (ng·mL^−1^)
Mangiferin	y = 37119x − 13.767	0.9998	3.74–22.44	26.7	93.5
Embinin	y = 25969x − 13.174	0.9999	4.40–198.00	11.5	16.5
Irislactin C	y = 19575x − 1.0423	0.9999	2.21–100.00	3.9	8.3
Irislactin A	y = 21296x + 0.0506	0.9998	2.52–113.40	16.7	31.5
Embinin A	y = 33469x − 3.9068	0.9999	3.36–37.10	23.6	84.0
Embinin C	y = 25250x − 2.1531	0.9999	4.00–180.00	8.9	15.0

**Table 3 molecules-23-03359-t003:** Contents of six components in *I. lactea* leaves from different regions (Mean ± SD, mg/g, *n* = 3).

No.	Mangiferin	Embinin	Irislactin C	Irislactin A	Embinin A	Embinin C
S1	1.60 ± 0.01	1.14 ± 0.01	1.68 ± 0.01	0.88 ± 0.01	4.79 ± 0.03	6.74 ± 0.08
S2	1.52 ± 0.01	2.14 ± 0.01	2.46 ± 0.01	1.64 ± 0.00	2.66 ± 0.02	5.26 ± 0.02
S3	1.71 ± 0.00	2.78 ± 0.01	0.90 ± 0.01	2.57 ± 0.01	2.15 ± 0.01	12.38 ± 0.02
S4	1.24 ± 0.02	1.39 ± 0.01	2.15 ± 0.03	1.16 ± 0.01	3.60 ± 0.05	6.29 ± 0.06
S5	1.72 ± 0.01	0.91 ± 0.00	4.31 ± 0.04	0.85 ± 0.01	6.09 ± 0.03	2.90 ± 0.00
S6	1.80 ± 0.01	3.75 ± 0.01	3.20 ± 0.02	3.11 ± 0.01	3.41 ± 0.03	6.32 ± 0.02
S7	0.69 ± 0.01	5.39 ± 0.01	0.75 ± 0.00	2.12 ± 0.01	0.92 ± 0.01	6.21 ± 0.07
S8	1.23 ± 0.02	2.85 ± 0.04	1.67 ± 0.02	2.43 ± 0.02	2.99 ± 0.02	7.80 ± 0.10
S9	1.16 ± 0.01	2.19 ± 0.03	1.95 ± 0.02	1.68 ± 0.01	2.80 ± 0.03	5.66 ± 0.01
S10	0.48 ± 0.01	11.78 ± 0.03	4.57 ± 0.03	2.42 ± 0.07	1.36 ± 0.03	2.30 ± 0.01
S11	1.31 ± 0.05	2.75 ± 0.01	2.66 ± 0.00	1.44 ± 0.02	4.23 ± 0.05	5.60 ± 0.02
S12	1.01 ± 0.01	2.29 ± 0.02	2.22 ± 0.01	1.39 ± 0.00	4.54 ± 0.02	8.08 ± 0.03
S13	1.72 ± 0.03	0.88 ± 0.01	4.43 ± 0.06	0.86 ± 0.02	6.67 ± 0.05	3.14 ± 0.01
S14	2.16 ± 0.01	2.59 ± 0.03	5.56 ± 0.02	0.77 ± 0.01	1.68 ± 0.03	0.49 ± 0.02

**Table 4 molecules-23-03359-t004:** Information for the investigated samples.

No.	Habitat	Collection	Collection Time	No.	Habitat	Collection	Collection Time
S1	Jiangsu	Nanjing	2015.04	S8	Shaanxi	Xi’an	2015.05
S2	Jiangsu	Nanjing	2015.04	S9	Beijing	Dongcheng	2015.05
S3	Jiangsu	Nanjing	2015.05	S10	Beijing	Haidian	2015.05
S4	Henan	Zhengzhou	2015.04	S11	Shandong	Zaozhuang	2015.05
S5	Tianjin	Jixian	2015.04	S12	Shandong	Zaozhuang	2015.05
S6	Tianjin	Tianjin	2015.04	S13	Liaoning	Huludao	2015.05
S7	Shanghai	Shanghai	2015.04	S14	Liaoning	Chaoyang	2014.09

All samples dried in the sun.

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
