# Peer review of "Qualitative and Quantitative Analysis of C-glycosyl-flavones of Iris lactea Leaves by Liquid Chromatography/Tandem Mass Spectrometry"

_molecules, 2018, doi:10.3390/molecules23123359_

Reviewer 1 Report

Dear authors,

The manuscript entitled "Qualitative and quantitative analysis of C-glycosylflavones of Iris lactea leaves by liquid chromatography tandem mass spectrometry" describes  the pharmaceutical value and quality, constituents of I. lactea leaves were determined by HPLC equipped with quadrupole time-of-flight tandem mass spectrometry. Twenty-two C-glycosylflavones were identified or tentatively identified. It presents scientific relevance for the area of Medicinal Chemistry and clinical and toxicology areas.

The language (English) are satisfactory (I suggest the final revision)! However, you need to change some details/informations in the Introduction, material and methods, results and discussion and conclusions. I request information on the procedures and interpretation of the results obtained.

1. Introduction section

- I suggest discussing more about C-glycosylflavones.

- Line 39: Reference 10 discusses the use of I. lactea as a type of pasture in the absence of winter fodder?

2.  Results

- Improve the quality of figure 1. What standards are used?

I suggest writing the "Results" and "Discussion" sections in the same item. The text (pages 5-6, lines 3-49) is well written and characterized as a discussion because the results obtained are compared with other authors.

- Page 6 (lines 58-59): Figure S1c shows that the extraction efficiency with methanol and ethanol are close. Why not use ethanol? Justify the choice of solvent.

- Page 6 (line 66): The best extraction times are in the range of 30 - 45 min .... Why did you select 15 min?

- Page 7 (lines 103-109): What are the concentration ranges for the reference compounds? How were LOD and LOQ calculated? Express your results in µg g-1 or µg mg-1! In Table 2, how were the linear working ranges defined? Refer to Fig. S3.

- Page 7 (lines 121, 125 and 126 and Fig 3.): To change "mg/g" by "mg g-1" in all manuscript.

- Page 7 (line 124): eight or six main components?

3.  Discussion

I suggest writing the "Results" and "Discussion" sections in the same item.

- Page 9 (lines 139-154): I suggest expanding the discussions.

4. Materials and Methods

- The samples were collected in 2015? How were they stored until use in the experiments? Were they lyophilized? Why was S6 sample selected for the experiments?

- Page 10 (lines 180-181): Did you only use mobile phase B in gradient elution program?

- Page 10 (line 183): To change "µl" by "µL" in all manuscript.

- Page 10 (line 186): To change "l/min" by "L min-1" in all manuscript.

- Page 10 (lines 190-191): The expression "The extraction conditions follow" is incomplete. What conditions?

- Page 11 (lines 209-215): I suggest rewriting in more detail: what does "the mixed standard solutions" mean? Nine concentrations? Which are? The authors report: “The mixed standard solution was diluted by methanol to determine the LOD and LOQ”…. How were LOD and LOQ calculated?

- Page 11 (lines 216-222): robustness was evaluated?

- Page 11 (line 225): what are the optimal parameters?

5. Conclusions: the article has no conclusions?

6. Tables and Figures: Ok! Improved quality of some figures and review units of concentration.

7. References: Please, check if the references are in accordance with the journal's rules.

Author Response

Responses to reviewer 1:

Dear reviewer,

We deeply appreciate your comments on our manuscript. The following are our responses.

1.      Introduction section

Comments 1: Authors are supposed to discuss more about C-glycosylflavones.

Response: Thank you for your suggestion. We have added the more discussion about C-glycosylflavones in our revised manuscript.

Comments 2: Reference 10 discusses the use of I. lactea as a type of pasture in the absence of winter fodder?

Response: Thank you for your suggestion. There is not the discussion about the use of I. lactea as a type of pasture in the absence of winter fodder in reference 10, the reference referred to the relevant content.  That is our negligence, we have added the relevant reference in our revised manuscript.

2.      Results

Comments 3: Improve the quality of Figure 1. What standards are used?

Response: Thank you for your suggestion. it is our negligence. We have improved the quality of Figure 1. Besides, standards of mangiferin, embinin, irislactin C, irislactin A, embinin A and embinin C corresponding to their numbered peaks in both chromatograms have been added to the remarks in Figure 1.

Comments 4: “I suggest writing the "Results" and "Discussion" sections in the same item. The text (pages 5-6, lines 3-49) is well written and characterized as a discussion because the results obtained are compared with other authors.”

Response: Thank you for your suggestion. The "Results" and "Discussion" sections have been written in the same item, and the other content have been revised according to your suggestion.

Comments 5: Figure S1c shows that the extraction efficiency with methanol and ethanol are close. Why not use ethanol? Justify the choice of solvent.

Response: Thank you for your suggestion, We chose methanol as the solvent for two reasons. Firstly, the extraction efficiency of methanol is slightly better than that of ethanol. Secondly, we found that the sample was directly injected after sample extraction, and the methanol extract was better separation effect than ethanol extract our liquid chromatography. So we chose methanol as the solvent in our experiment.

Comments 6: “The best extraction times are in the range of 30 - 45 min .... Why did you select 15 min?”

Response: Thank you for your suggestion. In our experiment, we found the extraction efficiency flattened out gradually after 30 minutes (extraction time). Therefore we choose the extraction time of 30 minutes as the intermediate point, and the range of 15–45 min was selected as the level of response surface design.

Comments 7: What are the concentration ranges for the reference compounds? How were LOD and LOQ calculated? Express your results in µg g-1 or µg mg-1! In Table 2, how were the linear working ranges defined? Refer to Fig. S3.

Response: Thank you for your comments. Concentration ranges for the reference compounds are linear ranges which is shown in Tab 2.; To determine the limits of detection and quantification, the mixed standard solutions containing the 6 reference compounds was diluted into a series of appropriate concentrations with methanol, and aliquots of the diluted solutions were injected into HPLC for analysis. The limits of detection (LOD) and quantification (LOQ) under the present chromatographic conditions were determined at S/N (the ratios of signal to noise) of 3 and 10, respectively; "µg/g" and "µg/mg" have been changed to "µg·g-1" and "µg·mg-1" in our revised manuscript. The linear working ranges were defined according to the minimum and maximum sample concentration and the lowest and highest values of the standard curve.

Comments 8: To change "mg/g" by "mg·g-1" in all manuscript.

Response: Thank you for your suggestion. "mg/g" has been changed to  "mg·g-1" in all manuscript.

Comments 9: eight or six main components?

Response: I am sorry. This is our clerical error, we have revised this error.

3.      Discussion

Comments 10: “I suggest writing the "Results" and "Discussion" sections in the same item.”

Response: Thank you for your suggestion. The "Results" and "Discussion" sections have been written in the same item.

Comments 11: “ suggest expanding the discussions.”

Response: Thanks for your suggestion. The discussions have been expanded.

4.      Materials and Methods

Comments 12: The samples were collected in 2015? How were they stored until use in the experiments? Were they lyophilized? Why was S6 sample selected for the experiments?

Response: thank you for your suggestion. The samples were collected in 2015, and our experiments were completed in 2016, For some reasons, we submit the manuscript now; All samples were dried in the sun; The quality of sample 6 is more than other samples, which can meet the requirements of the experiment, and the chromatogram is good.

Comments 13: “Did you only use mobile phase B in gradient elution program?”

Response: Sorry, we didn't express that clearly. We have revised this mistake.

Comments 14: To change "µl" by "µL" in all manuscript.

Response: Thank you for your advice. "µl" has been changed to "µL" in all manuscript.

Comments 15: To change "l/min" by "L·min-1" in all manuscript

Response: Thanks for your suggestion. "l/min" has been changed to "L·min-1" in all manuscript.

Comments 16: The expression "The extraction conditions follow" is incomplete. What conditions?

Response: Thank you for this comment on our manuscript. This is our clerical error, we have revised this error.

Comments 17: I suggest rewriting in more detail: what does "the mixed standard solutions" mean? Nine concentrations? Which are? The authors report: “The mixed standard solution was diluted by methanol to determine the LOD and LOQ”…. How were LOD and LOQ calculated?

Response: Thank you for your suggestion. we selected nine concentrations for the standard curve in our experiment, but only seven concentrations were selected for some standard products. As a result, there were nine points or seven points in the standard curve, which was the deficiency of our experiment. Therefore, we deleted the standard curve in the supporting materials; The LOD and LOQ: To determine the limits of detection and quantification, the mixed standard solutions containing the 6 reference compounds was diluted into a series of appropriate concentrations with methanol, and aliquots of the diluted solutions were injected into HPLC for analysis. The limits of detection (LOD) and quantification (LOQ) under the present chromatographic conditions were determined at S/N (the ratios of signal to noise) of 3 and 10, respectively.

Comments 18: robustness was evaluated?

Response: Thank you for your suggestion. Yes, the stability was evaluated in our manuscript.

Comments 19: what are the optimal parameters?

Response: Sorry, we didn't express that clearly. They are optimum extract parameters, which were obtained by single factor investigations combined with response surface methodology. We have revised this error.

5.      Conclusions

Comments 20: the article has no conclusions?

Response: Thank you for this comment on our manuscript. According to the journal's rules “conclusions section is not mandatory, but can be added to the manuscript if the discussion is unusually long or complex.”, as our discussion is short, so we wrote the "Conclusions" and "Discussion" sections in the same item. But we thought your suggestion of writing the "Results" and "Discussion" sections in the same item was more suitable, we have added conclusions section on our manuscript.

6.      Tables and Figures

Comments 21: Ok! Improved quality of some figures and review units of concentration.

Response: Thank you for your comment. We have revised our manscript.

7.      References

Comments 22: Please, check if the references are in accordance with the journal's rules.

Response: Thank you for reminding us. The DOI of these references has been added on our manuscript.

Thank you so much for all comments on our manuscript. The above are our answers. Please let us know if our answers still need to improve, we will try our best to fix these problems.

Reviewer 2 Report

The manuscript "Qualitative and quantitative analysis of C-glycosylflavones of Iris

lactea leaves by liquid chromatography tandem mass spectrometry" fits the scope of the journal. There are some remarks that I would like to address in order to improve the manuscript.

 The biological activity relevance of the determined compounds should be more clearly presented in section Discussion. The section Introduction should also be supplemented regarding the biological importance of the compounds which were found in larger quantities.

The aim of the article is not well defined and should also be amended.

The material and method section should be amended to be more accurate and in detail (especially the samples preparation and the optimization process).

The parameter "total peak area of target components" (rows 55-56, 195-196) should be explained in more detail as is a critical data for the optimization process.

Insufficient data are presented for Single factor experiments. A statistical analysis of the data to support the selection of the parameters to be optimized or an extensively discussion should be provided. For example, no significant differences are between the graphical results presented by the authors for the extraction moments (15, 30, etc) and the methods of extraction.

Fig 2 quality is very poor and should be improved.

The style should be polished. For example the plant name is written with lowercase in the description of table 3; rows 165-166:  "was prepared by dissolving powder of I. lactea leaves ... in 20 ml of 70% methanol" - dissolving should be change to  extracted.

Author Response

Responses to reviewer 2:

Dear reviewer,

We deeply appreciate your comments on our manuscript. The following are our responses.

Comments 1: The biological activity relevance of the determined compounds should be more clearly presented in section Discussion. The section Introduction should also be supplemented regarding the biological importance of the compounds which were found in larger quantities.

Response: Thank you for your suggestion. We have revised the relevant content in our revised manuscript.

Comments 2: The aim of the article is not well defined and should also be amended.

Response: Thank you for your suggestion. We have revised the aim of the article in our revised manuscript.

Comments 3: The material and method section should be amended to be more accurate and in detail (especially the samples preparation and the optimization process).

Response: Thank you for your suggestion. We have amended the samples preparation and the optimization process in our revised manuscript.

Comments 4: The parameter "total peak area of target components" (rows 55-56, 195-196) should be explained in more detail as is a critical data for the optimization process.

Response: Thank you for your suggestion. We have amended the samples preparation and the optimization process in our revised manuscript.

Comments 5: Insufficient data are presented for Single factor experiments. A statistical analysis of the data to support the selection of the parameters to be optimized or an extensively discussion should be provided. For example, no significant differences are between the graphical results presented by the authors for the extraction moments (15, 30, etc) and the methods of extraction.

Response: thank you for your suggestion. We have revised relevant content in revised manuscript.

Comments 6: Fig 2 quality is very poor and should be improved.

Response: Thank you for your suggestion. We have improved the Fig 2 in our revised manuscript.

Comments 7: The style should be polished. For example the plant name is written with lowercase in the description of table 3; rows 165-166:  "was prepared by dissolving powder of I. lactea leaves ... in 20 ml of 70% methanol" - dissolving should be change to extracted.

Response: Thank you for your comments. We have revised these errors.

Reviewer 3 Report

General comments:

Introduction: clear and pertinent (enough).

Fig 1. the graphical quality could be better.

Fig 2:   Respectfully, Figure 2 looks to me as a puzzle of 500 pieces of a white bear hidden in the snow. In my opinion, it is impossible to have a chemical clarity about the 22 species identified. This figure requires a power of abstraction well above average.

Is it really necessary?

I suggest to present it in the supplementary material.

Tab 2.  How you explain negative values for independent term of all compounds except for  Irislactin A  ? the Y axis is not an Area value ?

Why you used specifically these reported validation parameters (Linearity and limits of detection (LOD) and quantitation (LOQ), Precision, repeatability, stability and recovery) ?

 Why you used these reported validation parameters (Linearity and limits of detection (LOD) and quantitation (LOQ), Precision, repeatability, stability and recovery) ?

Page 8 lines 131-134: You wrote: The reason for the differences in total contents is likely such factors as different geographical conditions (GPS sampling location is welcome), harvesting time (when ? phenological information are welcome), processing technique (for example ?) and storage procedures (temperature ? O2 free ? be more explicit etc). 

I’m Sorry, but, what you know really about your samples (leaves) ? 

You have a very nice analytical procedure, but the information yours samples are very poor .

"In addition, some compounds were not tested (why ?) and so different compounds may convert to others (for example ?), which may alter some apparent concentrations".

What I can conclude about that? 

Are your results reliable based on these comments? 

I’m really Sorry for the "acid" question, but I don't know, what I can say about these two sentences.

I would like to see, if possible, an overlay of  Chromatograms comparing your  standards and your exctracts.

Did you a clean up (SPE for example) your samples before injection ?

 Fig 3… Perhaps a location map with  your samples could be interesting?

Fig. 3 Total contents of ”main” components in Iris lactea leaves from different areas

 Why you use the term "main" ?  or in other words…define “main” ?

Why these 6 compounds ? (Using the chromatograms (how ?), the six main components of I. lactea leaves from different regions were quantitatively analyzed.

Statistical tests can be helpfull  in your interpretation of tab 3 and fig 3.

Discussion: I am Sorry, but it is very generic and present a very modest argumentation.

 Materials and methods

Chemicals and plant material

How you know if you work with the same plant specie ?

Have you done an ethnobotanical or taxonomic charactetization ? 

Have you a "voucher" or "deposit numbers" your plants in a biotery ?

I am talking about rastreability. Geostatistic distribution, etc.  

Tab 4: where specifically you take your samples ?

Page 10 – line 167 two ultrasonic extractions... be more specific and present the conditions (potency ? etc).

Qualitative analysis of HPLC-Q-TOF-MS/MS for chemical constituents of I. lactea leaves

How you come to this condition ?  "Mobile phases A (0.1% formic

181 acid–water) and B (acetonitrile) with gradient elution program were set as follows: 0 min, 5% B; 5–10 182 min, 11% B; 15 min, 19% B; 20–24 min, 24% B; 25 min, 27% B; 28 min, 35% B; 30 min, 38% B; 35 min, 70% B and 40–45 min, 95% B".

Why 30° C ? and how/where you found these experimental conditions ?

4.4 Optimization of the extraction process

4.4.1 Single factor experiments

Ultrasonic conditions ?  why 30 min ? why reflux 80°C / 1h ?

Box–Behnken response-surface design experimente  (Why this model ?)

 ************************ 

Supplementary material

Fig s1

a.       Why you used a mesh of 80 if the larger peak area was for a mesh 100 ?

b.      What is “merceration” ?

c.       Use "log" scale in total peak axis. 

It is better to observe the differencies between each solvent combination.

Fig 3 is not necessary in my opinion.

Tab S1 (can you use a multivariate analysis to help you in the interpretation your model,  like a PCA (principal compound analysis) for example ?

Tab S2

For the number 3.255E-003 and in thoer cases... the notation “E-“ is  correct ?

Tabs S4 to S9 are your results of the merit figres, but they were not discussed or explored in your text. 

Final comments: this manuscript generically presents a good analytical description. I had some doubts reading this work and I think the authors should be more argumentative (more citations are welcome) and pay more attention to the discussion.

Author Response

Responses to reviewer 3:

Dear reviewer,

We deeply appreciate your comments on our manuscript. The following are our responses.

1.      Introduction

Comments 1: clear and pertinent (enough).

Response: Thank you for your comments. We have revised the introduction in our revised manuscript.

2.      Results

Comments 2: The graphical quality of Fig 1 could be better.

Response: Thank you for your comments. it is our negligence. We have improved the graphical quality of Figure 1.

Comments 3: Fig 2: Respectfully, Figure 2 looks to me as a puzzle of 500 pieces of a white bear hidden in the snow. In my opinion, it is impossible to have a chemical clarity about the 22 species identified. This figure requires a power of abstraction well above average. Is it really necessary? I suggest to present it in the supplementary material.

Response: thank you for your suggestion, we identified or tentatively identified the structure of compounds according to our previous studies, which were reported about the isolation and identification of chemical component in Iris lactea leaves. In our previous studies, a series of C-glycosylflavones were isolated from Iris lactea leaves. Therefore we identified or tentatively identified the structure of compounds according to the previous isolated compounds in the study. Of course, some compounds (A12, A13, A17 and A19) are not identified by previous studies, It is our fault. We have revised relevant content in our revised manuscript, in addition, we have moved figure 3 to the supporting material according to your suggestion, thank you very much.

Comments 4: Tab 2. How you explain negative values for independent term of all compounds except for Irislactin A ? The Y axis is not an Area value ?

Response: thank you for your suggestion, That's allowed that the intercept of Y axis may be positive or negative values in the standard curve, and the absolute values of all the curves are within the allowable range. For the reason of the negative values for independent term of all compounds except for Irislactin A, It may be the result of the instrument or the coincidence etc, it's hard to explain the reason, but we think it's reasonable, The Y axis is an area value.

Comments 5: Why you used specifically these reported validation parameters (Linearity and limits of detection (LOD) and quantitation (LOQ), Precision, repeatability, stability and recovery)?

Response: Thank you for your suggestion, we used validation parameters (Linearity and limits of detection (LOD) and quantitation (LOQ), Precision, repeatability, stability and recovery) according to the previous literatures and our studies.

Comments 6: Why you used these reported validation parameters (Linearity and limits of detection (LOD) and quantitation (LOQ), Precision, repeatability, stability and recovery)?

Response: Thank you for your suggestion, we used validation parameters (Linearity and limits of detection (LOD) and quantitation (LOQ), Precision, repeatability, stability and recovery) according to the previous literatures and our studies.

Comments 7: Page 8 lines 131-134: You wrote: The reason for the differences in total contents is likely such factors as different geographical conditions (GPS sampling location is welcome), harvesting time (when? phenological information are welcome), processing technique (for example?) and storage procedures (temperature? O2 free? be more explicit etc).

Response: Thank you for your suggestion, we have revised the relevant content in our revised manuscript. In addition, processing technique includes post-mining drying mode, drying temperature and so on,The storage procedures includes temperature, humidity and air circulation of storage etc.

Comments 8: I’m Sorry, but, what you know really about your samples (leaves)? You have a very nice analytical procedure, but the information yours samples are very poor.

Response: thank you for your suggestion. that all the medicinal materials are Iris lactea leaves, and these samples were collected by us personally or by professionals from 2014 to 2015, and our experiments were completed in 2016, For some reasons, we submit the manuscript now.

Comments 9: "In addition, some compounds were not tested (why ?) and so different compounds may convert to others (for example ?), which may alter some apparent concentrations".

Response: Thank you for your suggestion, some compounds have not been tested because we did not get the corresponding standard by the previous experiment or purchase. In previous isolation and identification experiments, we found that some compounds had mutual transitions, such as compounds A19 and A22.

Comments 10: What I can conclude about that?

Are your results reliable based on these comments?

I’m really sorry for the "acid" question, but I don't know, what I can say about these two sentences.

Response: Thank you for your suggestion, it is our writing problem, we have revised the relevant content in our revised manuscript.

Comments 11: I would like to see, if possible, an overlay of Chromatograms comparing your standards and your exctracts.

Response: Thank you for your suggestion, we have revised the content in the revised manuscript.

Comments 12: Did you a clean up (SPE for example) your samples before injection ?

Response: Thank you for your suggestion, the samples were filtered through a 0.22 μm membrane filter before injection, and the results were satisfactory in chromatogram. But we haven't done a clean up (SPE for example) for our samples before injection, this is a flaw.

Comments 13: Fig 3… Perhaps a location map with your samples could be interesting?

Response: Thank you for your suggestion, We tried to combine location map with our samples, but the results were not satisfactory, so we gave up.

Comments 14: Fig. 3 Total contents of “main” components in Iris lactea leaves from different areas. Why you use the term "main"? or in other words…define “main” ?

Why these 6 compounds? (Using the chromatograms (how?), the six main components of I. lactea leaves from different regions were quantitatively analyzed.

Response: Thank you for your suggestion. From the chromatogram, we could find that compounds A3, A11, A15, A18, A19, and A22 are the main ingredients in the Iris lactea leaf, so we use this term. We obtained the standard products of six compounds by previous experiments and purchases, and six compounds are the main ingredients of I. lactea leaves, so we quantified these 6 ingredients.

Comments 15: Statistical tests can be helpful in your interpretation of tab 3 and fig 3.

Response: Thank you for your suggestion, We tried to use the hierarchical clustering analysis (HCA) and principalcomponent analysis (PCA) and so on for analysis the results, but the results were not satisfactory, so we gave up.

3.      Discussion:

Comments 16: I am Sorry, but it is very generic and present a very modest argumentation.

Response: Thank you for your comment, we have revised the discussion in revised manuscript.

4. Materials and methods

Chemicals and plant material

Comments 17: How you know if you work with the same plant specie ?

Response: Sorry, it is our negligence. Leaves of I. lactea from different area were identified by prof. Minjian Qin, China Pharmaceutical University, Nanjing, China. We have added it in our manuscript.

Comments 18: Have you done an ethnobotanical or taxonomic charactetization ?

Response: Thank you for your comment, we've done taxonomic work before, so we'll write taxonomic significance in the discussion

Comments 19: Have you a "voucher" or "deposit numbers" your plants in a biotery? I am talking about rastreability. Geostatistic distribution, etc. 

Response: Yes, the specimens (No. ITM20150516-ITM20150519) were deposited in the Herbarium of Medicinal Plants of China Pharmaceutical University. It is our negligence, We have added it in our manuscript.

Comments 20: Tab 4: where specifically you take your samples?

Response: Yes, Table 4 includes the samples information.

Comments 21: Page 10 – line 167 two ultrasonic extractions... be more specific and present the conditions (potency? etc).

Response: Sorry, it is our negligence. The conditions of ultrasonic were at 25 °C, 100 W for 45 min. We have added them in our manuscript.

Qualitative analysis of HPLC-Q-TOF-MS/MS for chemical constituents of I. lactea leaves

Comments 22: How you come to this condition? "Mobile phases A (0.1% formic acid–water) and B (acetonitrile) with gradient elution program were set as follows: 0 min, 5% B; 5–10 182 min, 11% B; 15 min, 19% B; 20–24 min, 24% B; 25 min, 27% B; 28 min, 35% B; 30 min, 38% B; 35 min, 70% B and 40–45 min, 95% B". Why 30° C? and how/where you found these experimental conditions ?

Response: Thank you for your comment. We had investigated and optimized the chromatographic conditions before we came to this condition. And this part we have added to our manuscript.

4.4 Optimization of the extraction process

4.4.1 Single factor experiments

Comments 23: Ultrasonic conditions? why 30 min ? why reflux 80°C / 1h ?

Response: Thanks for your comment. As we consulted some literatures and considered our laboratory equipment conditions, three extraction method and its specific condition were selected.

Comments 24: Box–Behnken response-surface design experiment (Why this model?)

Response: Thanks for your comment. In previous studies, we found that Box–Behnken response-surface design experiment possessed a better effect for investigation of extraction factors , so we used it.

Supplementary material

Fig S1

Comments 25: a. Why you used a mesh of 80 if the larger peak area was for a mesh 100 ?

Response: Thanks for your comment. We have preliminary experiment before that there is no significant difference used a mesh of 80 or 100. In addition, Iris lactea leaves are highly fibrous, the crushed samples cannot pass through the sieve very good if a mesh 100.

Comments 26: b. What is “merceration” ?

Response: I am sorry. This is our clerical error, we have revised this error.

Comments 27: c. Use "log" scale in total peak axis.

Response: Thanks for your comment. We have revised the relevant content in revised manuscript.

Comments 28: It is better to observe the differencies between each solvent combination.

Response: Thank you for your suggestion. Our experiment was ill-conceived. In future experiments it will be improved and perfected.

Comments 29: Fig S3 is not necessary in my opinion.

Response: Thank you for your suggestion. We removed Fig S3 according to your suggestion.

Comments 30: Tab S1 (can you use a multivariate analysis to help you in the interpretation your model, like a PCA (principal compound analysis) for example?

Response: Thank you for your suggestion. Tab S1 shows the results derived directly from Box–Behnken response-surface design experiment.

Comments 31: Tab S2 For the number 3.255E-003 and in thoer cases... the notation “E-” is correct?

Response: Thank you for your suggestion. We have revised the error in revised manuscript.

Comments 32: Tabs S4 to S9 are your results of the merit figres, but they were not discussed or explored in your text.

Response: Thank you for your suggestion. Tabs S4 to S9 Table 4 are the results of methodological verification, and they are within the allowable range, so we describe them briefly in the article.

Final comments: this manuscript generically presents a good analytical description. I had some doubts reading this work and I think the authors should be more argumentative (more citations are welcome) and pay more attention to the discussion.

Thank you so much for all comments on our manuscript. The above is our answer. Please let us know if our answers still need to improve, we will try our best to fix these problems.

Reviewer 4 Report

Manuscript from Qin Minjian Xie Guoyong and colleagues describes a chromatography-based analysis on Iris lactea leaves. The reported results are interesting, but there are some severe problems in the paper that have to be solved to make it suitable for publication.

1) Data reported in table 1 are confused and there are some incongruences: m/z calculated in some cases are related to the reported molecular formula (i.e. A9, A10…) whereas in other cases are related to the observed adduct (i.e. A17, A19…). Moreover, it is not clear what’s the  difference between [M+HCOO]- and [M+HCOOH-H]- ions.

2) It is well known that during the ESI process some adduct formation can occur, such as addition of Na+ or HCOO- ions to the compounds, thus producing cationized [M+Na]+ or anionized [M+HCOO]- ions. However, if during fragmentation the adduct breaks down, the compound returns to be neutral [M] and, therefore, invisible for the mass spectrometer. On these basis I think that MS/MS data reported for compound A9 are not in agreement with the reported structure

3) In my opinion it is quite “dangerous” to define the sugars on the mere basis of MS and MS/MS data, since it is hard to distinguish between different hexoses. Although the authors used some standards, in the absence of NMR data, I think that it is more correct to use less specific definitions.

4) The starting points of the linear ranges reported in table 2 are lower than the measured LOQs. How is it possible?

5) Methods are inadequately described

Author Response

Responses to reviewer 4:

Dear reviewer,

We deeply appreciate your comments on our manuscript. The following are our responses.

Comments 1: Data reported in table 1 are confused and there are some incongruences: m/z calculated in some cases are related to the reported molecular formula (i.e. A9, A10…) whereas in other cases are related to the observed adduct (i.e. A17, A19…). Moreover, it is not clear what’s the difference between [M+HCOO]- and [M+HCOOH-H]- ions.

Response: Sorry, it is our negligence. We have revised this error.

Comments 2: It is well known that during the ESI process some adduct formation can occur, such as addition of Na+ or HCOO- ions to the compounds, thus producing cationized [M+Na]+ or anionized [M+HCOO]- ions. However, if during fragmentation the adduct breaks down, the compound returns to be neutral [M] and, therefore, invisible for the mass spectrometer. On these basis I think that MS/MS data reported for compound A9 are not in agreement with the reported structure.

Response: thank you for your suggestion, it is our negligence, we have revised the error in revised manuscript.

Comments 3: In my opinion it is quite “dangerous” to define the sugars on the mere basis of MS and MS/MS data, since it is hard to distinguish between different hexoses. Although the authors used some standards, in the absence of NMR data, I think that it is more correct to use less specific definitions.

Response: thank you for your suggestion, we identified or tentatively identified the structure of compounds according to our previous studies, which were reported about the isolation and identification of chemical component in Iris lactea leaves. In our previous studies, a series of C-glycosylflavones were isolated from Iris lactea leaves by NMR, MS, IR etc. Therefore we identified or tentatively identified the structure of compounds according to the previous isolated compounds and MS data in the study. Of course, we corrected some errors and unreasonable structure according to your suggestion.

Comments 4: The starting points of the linear ranges reported in table 2 are lower than the measured LOQs. How is it possible?

Response: I am sorry. This is our clerical error. The units of LOQs and LODs should be “ng·mL-1”. We have revised this error.

Comments 5: Methods are inadequately described.

Response: Thank you for your comment. We have described methods in more details.

Thank you so much for all comments on our manuscript. The above are our answers. Please let us know if our answers still need to improve, we will try our best to fix these problems.

Reviewer 5 Report

Authors present an interesting study entitled “Qualitative and quantitative analysis of C3 glycosylflavones of Iris lactea leaves by liquid 4 chromatography tandem mass spectrometry” The article is well structured and written. However, the authors should improve some aspect.:

Pag 2 The figure 1 is unreadable, the authors should improve the resolution.

Pag 6 the author should enrich the reference because it is very incomplete:

Row 60:

D'Archivio A.A., Maggi M.A., Ruggieri F., Carlucci M., Ferrone V., Carlucci G., Optimisation by response surface methodology of microextraction by packed sorbent of non steroidal anti-inflammatory drugs and ultra-high performance liquid chromatography analysis of dialyzed samples Journal of Pharmaceutical and Biomedical Analysis Volume 125, June 05, 2016, Pages 114-121

Berger-Brito, I., Machour, N., Morin, C., Portet-Koltalo, F. Experimental Designs for Optimizing Multi-residual Microwave-assisted Extraction and Chromatographic Analysis of Oxygenated (Hydroxylated, Quinones) Metabolites of PAHs in Sediments Chromatographia Volume 81, 2018, Pages 1401-1412

Row 69:

D'Archivio, A.A., Maggi, M.A. Investigation by response surface methodology of the combined effect of pH and composition of water-methanol mixtures on the stability of curcuminoids Food Chemistry Volume 219, 2017, Pages 414-418

Pag 60 row56: The author should evaluate the effect of ultrasounds on stability of analyzed compounds

Pag 60 row 74:The coefficients of equation are different than the data reported in table S3….why?

In Table S3 are reported incorrect data like freedom degree and Standard Error (there are same values for any factor……)

Pag 7 Table 2 The authors should control the LOD and LOQ, they not are comparable with linear range of calibration curves

Tab3 authors should control the standard deviations of measure, there are too low and unreal values

Author Response

Responses to reviewer 5:

Dear reviewer,

We deeply appreciate your comments on our manuscript. The following are our responses.

Comments 1: Pag 2 The figure 1 is unreadable, the authors should improve the resolution.

Response: Sorry, it is our negligence. We have improved the quality of Figure 1.

Comments 2: Pag 6 the author should enrich the reference because it is very incomplete:

Row 60: D'Archivio A.A., Maggi M.A., Ruggieri F., Carlucci M., Ferrone V., Carlucci G., Optimisation by response surface methodology of microextraction by packed sorbent of non steroidal anti-inflammatory drugs and ultra-high performance liquid chromatography analysis of dialyzed samples Journal of Pharmaceutical and Biomedical Analysis Volume 125, June 05, 2016, Pages 114-121

Berger-Brito, I., Machour, N., Morin, C., Portet-Koltalo, F. Experimental Designs for Optimizing Multi-residual Microwave-assisted Extraction and Chromatographic Analysis of Oxygenated (Hydroxylated, Quinones) Metabolites of PAHs in Sediments Chromatographia Volume 81, 2018, Pages 1401-1412

Row 69: D'Archivio, A.A., Maggi, M.A. Investigation by response surface methodology of the combined effect of pH and composition of water-methanol mixtures on the stability of curcuminoids Food Chemistry Volume 219, 2017, Pages 414-418

Response: thank you for your suggestion, we added the reference in the corresponding place according to your suggestion.

Comments 3Pag 60 row56: The author should evaluate the effect of ultrasounds on stability of analyzed compounds

Response: thank you for your suggestion, In our preliminary experiments, we considered the effect of ultrasounds on stability of analyzed compounds,  which has no effect on the stability of the compound.

Comments 4Pag 60 row 74:The coefficients of equation are different than the data reported in table S3….why?

Response: thank you for your suggestion, we checked the relevant content and made modifications in revised manuscript.

Comments 5In Table S3 are reported incorrect data like freedom degree and Standard Error (there are same values for any factor……)

Response: thank you for your suggestion, we checked the relevant content again and compared with the literature, we verified that like freedom degree and Standard Error were correct.

Comments 6Pag 7 Table 2 The authors should control the LOD and LOQ, they not are comparable with linear range of calibration curves

Response: I am sorry. This is our clerical error. The units of LOQs and LODs should be “ng·mL-1”. We have revised this error.

Comments 7Tab3 authors should control the standard deviations of measure, there are too low and unreal values

Responses: thank you for your suggestion, we checked the relevant content and the standard deviations of measure were correct.

Thank you so much for all comments on our manuscript. The above are our answers. Please let us know if our answers still need to improve, we will try our best to fix these problems.

Round  2

Reviewer 1 Report

The authors answered the questions and improved the work from the suggestions.

I am fully satisfied with the changes!

Reviewer 3 Report

The authors  reviewed criteriously the entire manuscript. Congratulations!

Reviewer 4 Report

The authors significnatly improved the manuscript,satisfactorily responding to the criticisms made by the referees

Reviewer 5 Report

The manuscript is now suitable for publication